# OpenReview forum: "Modal Aphasia: Can Unified Multimodal Models Describe Images From Memory?"
_ICLR.cc/2026/Conference — ICLR 2026 Poster_

### Official Review · Reviewer_tN39 · 2025-10-28

**Soundness:** 3
**Presentation:** 3
**Contribution:** 3
**Rating:** 4
**Confidence:** 2

**Summary:**

This paper analyzes the phenomenon of modal aphasia, where unified multimodal LLMs models may be able to illustrate concepts that they fail to accurately describe in text. The paper systematically demonstrates the existence of this phenomenon in both proprietary and open-source multi-modal LLMs, as well as showing a consequence of this phenomenon in creating vulnerabilities in multi-modal LLMs aligned to prevent harmful outputs.

**Strengths:**

The concept of modal aphasia is intuitive and cleverly formulated, and well-motivated by connections to studies of visual and verbal cognition in humans and phenomena such as aphantasia.

It is an interesting and potentially surprising finding that unified multimodal LLMs taught new visual concepts fail to describe them accurately in text. It is clear how this is a drawback of these models and important to understand in order to mitigate such misalignments. The connection to work on visualizing concepts for reasoning is also interesting.

The experiments on open-weights models are appreciated for reproducible science.

Overall the paper is clear and well-written.

**Weaknesses:**

I find the motivation from frontier models (L41–47) and tests on GPT-5 (Sec 3) unconvincing, because it is likely that GPT-5 routes image generation requests to a separate image generation model rather than specifically training the LLM and image generator in parallel. The [GPT-5 system card](https://cdn.openai.com/gpt-5-system-card.pdf) only mentions GPT-5 accepting textual or image input, but noticeably does not discuss image generation.

It’s not obvious to me that performance when fine-tuning on a new visual concept would be comparable to zero-shot evaluation of the pretrained model’s knowledge of visual concepts. It could be that fine-tuning on a single concept alone degrades the LLM’s general reasoning abilities and differs from the dynamics of pretraining. Why not also perform a similar zero-shot test to that in Sec 3 applied to open models?

I’m not sure if the effect demonstrated in Sec 5 is due to modal aphasia. Modal aphasia was previously defined (L32-33) as the inability to access knowledge in text which can be expressed in images, while Sec 5 discusses the case where a visual concept can still be described in text using a circumlocution.

Overall, I think there is a valuable conceptual and methodological contribution here, and I'm willing to reconsider my score if the significant points above are addressed.

**Questions:**

The paper focuses on modal aphasia where a concept can be illustrated but not described in text. Have you considered the reverse phenomenon? (Models being able to describe concepts in words that they struggle to illustrate.) Or whether this could generalize to other modalities which are now being processed by unified multimodal LLMs?

There is some evidence that images may provide supervision for language models that are trained multimodally, reflected in visual reasoning in text, beyond what is learned from text alone. [1–3] On the other hand, this paper’s results suggest that textual description of visual concepts is not readily learned from text-image pairs. How should we interpret these findings in this context?

Does the effect also hold when the model’s visual parameters are trainable? (vs. L298)

Should “ChatGPT-5” be “GPT-5”?

Figs 4–5 are missing a visualization of the baseline (random) performance, making it harder to visually interpret the results.

[1] Zhang et al. Visual Commonsense in Pretrained Unimodal and Multimodal Models. NAACL 2022

[2] Liu et al. Things not Written in Text: Exploring Spatial Commonsense from Visual Signals. ACL 2022

[3] Alper et al. Is BERT Blind? Exploring the Effect of Vision-and-Language Pretraining on Visual Language Understanding. CVPR 2023

---

> ### Author Response · Authors · 2025-11-20
>
> We thank the reviewer for their thoughtful inputs and appreciate that they found our contribution interesting and surprising.
>
> > [W1] I find the motivation from frontier models (L41–47) and tests on GPT-5 (Sec 3) unconvincing, because it is likely that GPT-5 routes image generation requests to a separate image generation model
>
> We kindly refer the reviewer to our general response for a detailed answer. In short, even if the final image generation is indeed performed by a separate image decoder model, this is likely done based on representations that can be decoded both into images and text. Yet, unexpectedly, the text modality fails to access the knowledge in those representations.
>
> > [W2] It’s not obvious to me that performance when fine-tuning on a new visual concept would be comparable to zero-shot evaluation [...]. It could be that fine-tuning on a single concept alone degrades the LLM’s general reasoning abilities and differs from the dynamics of pretraining.
>
> We thank the reviewer for this useful suggestion. We did try to replicate the real-world movie poster experiments on the base Harmon/Janus-Pro models (zero-shot), but found that neither model can accurately reproduce any posters. We will mention this in the updated version of the paper.
>
> We also agree that fine-tuning does not perfectly replicate pretraining. Thus, we carefully pick hyperparameters that yield minimal degradation of general-purpose reasoning and image-generation capabilities (as indicated by the benchmark scores in Table 4).
>
> Moreover, as mentioned in our general response, a desirable feature of our fine-tuning setup is that it guarantees that memorization must occur in the model's shared representation space, and thus that any evidence of modal aphasia we discover cannot be trivially attributed to memorization in modality-specific components.
>
> > [W3] I’m not sure if the effect demonstrated in Sec 5 is due to modal aphasia. [...] Sec 5 discusses the case where a visual concept can still be described in text using a circumlocution.
>
> We argue in the general response how modal aphasia causes the "circumlocation": because the models do not "understand" the visual concept of feet, they only learn to refuse prompts mentioning the word "feet", instead of learning to avoid generating the concept of feet.
>
> > [Q1] Have you considered the reverse phenomenon? (Models being able to describe concepts in words that they struggle to illustrate.) Or whether this could generalize to other modalities which are now being processed by unified multimodal LLMs?
>
> This is a great question! There are indeed related failure modes for the reverse direction that have been mentioned in past work. A simple example is counting: GPT-5 can verbally express that the US flag contains 50 stars, but often includes a different number of stars in generated images.
>
> We also believe that modal aphasia generalizes to other modalities, such as audio, since nothing about the disassociation is specific to images.
> Our focus on images is because images are the second most common modality after text, and since most open-weight unified models do not support additional modalities.
>
> > [Q2] There is some evidence that images may provide supervision for language models that are trained multimodally [...].  How should we interpret these findings in this context?
>
> We thank the reviewer for the references and believe they are worth discussing in the updated version of the paper.
>
> While modal aphasia has implications for visual commonsense reasoning, there are important differences:
> Visual commonsense reasoning considers how well models *learn* a concept when *trained on text and images vs. text-only*, evaluating a *single modality*. Modal aphasia considers models that *always jointly train all modalities*, do learn concepts well, and compares how well models can *access/express* learned knowledge in *one modality vs. another*.
>
> In particular, the referenced works conclude that training on images and text improves reasoning over visual concepts; however, our experiments show that even when training on images and text, the text modality can fail to access knowledge that the image modality can.
>
> > [Q3] Does the effect also hold when the model’s visual parameters are trainable? (vs. L298)
>
> In short, when also training the visual parameters, we expect an even stronger disassociation (but it would be entangled with factors beyond modal aphasia). Training only the backbone LLM hence strengthens the robustness of our findings.
> We provide a more in-depth discussion in the general response.
>
> > [Q4] Should “ChatGPT-5” be “GPT-5”?
>
> We use the ChatGPT web interface with GPT-5 to be as realistic as possible. The updated version of the paper will clarify this.
>
> > [Q5] Figs 4–5 are missing a visualization of the baseline (random) performance, making it harder to visually interpret the results.
>
> We thank the reviewer for this suggestion and will add random-guessing baselines in Figures 4 and 5.

---

> ### Author Response · Authors · 2025-11-28
>
> Dear Reviewer tN39,
>
> We would like to kindly ask you to let us know whether our rebuttal resolved your weaknesses, and if so, to reflect this in your scores. Otherwise, we are happy to continue the discussion!
>
> To summarize the most important points:
>
> 1. We clarify how our findings are sound despite not knowing the specific internals of (Chat)GPT-5.
> 2. We cannot do zero-shot movie poster experiments with Janus-Pro or Harmon, because they did not sufficiently memorize any poster (even visually). But we ensured that our fine-tuned models remain close to the pretrained ones.
> 3. We explain that the "circumlocution" in our safety case-study is precisely caused by modal aphasia.
> 4. We believe that modal aphasia also applies to other modalities (e.g., audio) and that the reverse phenomenon also applies (with some limitations).
> 5. We added a discussion comparing modal aphasia to insights around visual/spatial commonsense reasoning.
> 6. If we also trained the visual parameters, we expect an even stronger disassociation (but it would be entangled with factors beyond modal aphasia).

---

### Official Review · Reviewer_Zj2z · 2025-10-31

**Soundness:** 2
**Presentation:** 3
**Contribution:** 2
**Rating:** 2
**Confidence:** 4

**Summary:**

The paper introduces the concept of modal aphasia, where unified multimodal models accurately generate images yet fail to generate the same knowledge through text. The authors find that this phenomenon persists across different model architectures and training procedures. They further highlight its implications for AI safety: safety interventions applied to one modality do not reliably transfer to another (e.g., a model aligned in text may still produce harmful images).

**Strengths:**

* The paper addresses a fundamental question in unified multimodal modeling: whether such models can express the same knowledge consistently across modalities.
* It also provides a comparative analysis of both proprietary and open-source models to support the generality of its claims.

**Weaknesses:**

* The experimental design is questionable. The authors train their unified multimodal models *only* on the image generation task—learning to produce images from text prompts. This setup naturally biases the model toward visual generation without ensuring it can express the same concepts in language, since it is not trained on captioning tasks. A more appropriate design would jointly train the model on both image generation and image captioning using the same image–text pairs, then compare its performance across modalities.

* The paper conflates modal aphasia with a prompt engineering loophole. Regarding safety risks (Section 5), the authors interpret reduced refusal on real words ("feet") vs. rare expression prompts ("secondary balance units") as evidence of modal aphasia. However, this may not necessarily be due to a modality gap. It may simply be that the text input pipeline is not robust to rephrasing / prompt engineering. Moreover, the experiment is based on a single example ("feet" vs. "secondary balance units"), making its findings too limited in scope to support a general claim.

* The assumption that modern unified architectures train image and language jointly "from scratch" is invalid. Section 3 reports initial experiments on ChatGPT, whose model details are undisclosed. For instance, the model may not be a single jointly trained vision-language architecture, but rather a combination of a separate vision-language model and an image generation model accessible through a unified chat interface. Furthermore, this statement does not hold for the two open-sourced models analyzed in Section 4, Janus-Pro and Harmon, neither of which is trained jointly from scratch. Harmon combines an image generation model (MAR) and LLM (Qwen2.5-1.5B-Instruct), which are trained separately on large-scale datasets describing distinct modalities and concepts that far exceed the scale of data used for vision–language alignment. The same applies to Janus-Pro, which combines separately trained components (DeepSeek-LLM and SigLIP). Consequently, the image and language components capture different visual and linguistic concepts, making modal aphasia an expected, not novel, outcome.

**Questions:**

The paper poses a very interesting question, but the experiments are poorly designed and lack generalizability. As mentioned in the weaknesses, it would be important to examine what happens when the models are trained on both image generation and captioning tasks using the same image–text pairs. Would the same trend persist under this training setup?

---

> ### Author Response · Authors · 2025-11-20
>
> We thank the reviewer for the critical assessment of our experiments and hope that our response provides more clarity. We answer their concerns in a different order, because we believe that our explanation for the third point also helps to resolve earlier ones.
>
> > [W3] The assumption that modern unified architectures train image and language jointly "from scratch" is invalid. Section 3 reports initial experiments on ChatGPT, whose model details are undisclosed. [...]. For instance, the model may not be a single jointly trained vision-language architecture, but rather a combination of a separate vision-language model and an image generation model [...]. Furthermore, this statement does not hold for the two open-sourced models analyzed in Section 4, Janus-Pro and Harmon, neither of which is trained jointly from scratch. [...] Consequently, the image and language components capture different visual and linguistic concepts, making modal aphasia an expected, not novel, outcome.
>
> We thank the reviewer for pointing out our incorrect use of "from scratch" on L46 and remove it in the updated version of the paper.
>
> Regarding their concerns about GPT-5's training, we kindly refer the reviewer to our general response. In summary, even if the final image generation is performed by some separate image decoder model, this is likely done based on representations that can be decoded both into images and text. Hence, the text modality has access to representations that capture the visual concepts, but unexpectedly fails to access the knowledge in those representations.
>
> For Harmon and Janus-Pro, we only consider models after fine-tuning on novel concepts; we do not show modal aphasia in the base models. Hence, *the way those models were pretrained does not affect our findings* (although both models train modalities jointly for many steps).
>
> In particular, we fine-tune only the backbone LLMs, so that any knowledge about the new concepts can only be memorized in the unfied representation space. This rules out that "image and language components capture different visual and linguistic concepts". Hence, by design, our experiments ensure that there is *never* any "expected" trivial version of modal aphasia.
>
> We are happy to elaborate more if the reviewer could tell us in which sense they think this setup makes our observations expected.
>
> > [W1] The authors train their unified multimodal models only on the image generation task [...]. This setup naturally biases the model toward visual generation without ensuring it can express the same concepts in language, [...]. A more appropriate design would jointly train the model on both image generation and image captioning [...].
>
> > [Q1] [...] it would be important to examine what happens when the models are trained on both image generation and captioning tasks using the same image–text pairs. Would the same trend persist under this training setup?
>
> As mentioned in the previous response, we train only the backbone LLM, so that all memorization of new concepts must happen in the unified representation space.
>
> Therefore, there is little difference between image generation and image captioning. In both cases, transformers act on a sequence of representations that encode concepts in a unified space. The same trends are expected for both training paradigms.
>
> We thus decided to fine-tune only for image generation, as the added complexity of training on multiple tasks does not meaningfully change the setup or conclusions.
>
> We hope this explanation helps to clarify the reviewer's concerns. If not, we are curious to hear in which sense they think mixing tasks would improve the soundness of our experiments.
>
> > [W2] Regarding safety risks (Section 5), the authors interpret reduced refusal on real words ("feet") vs. rare expression prompts ("secondary balance units") as evidence of modal aphasia. [...]. It may simply be that the text input pipeline is not robust to rephrasing / prompt engineering.
>
> We understand the reviewer's concerns. In the general response, we detail how we designed the experiments to be as robust against prompt engineering/rephrasing as possible.
>
> > [W2] Moreover, the experiment is based on a single example ("feet" vs. "secondary balance units"), making its findings too limited in scope to support a general claim.
>
> We do not make a general claim that the study directly reflects practice. Instead, our safety case-study is an illustrative example of how modal aphasia *might* affect a system's safeguards. Because real-world safeguards are complex (and often unknown), we deliberately keep the case-study simple for the sake of illustration. Crucially, the example is not cherry-picked or special, and we expect similar outcomes with different harmful concept types.

---

> ### Author Response · Authors · 2025-11-28
>
> Dear Reviewer Zj2z,
>
> We would like to kindly ask you to let us know whether our rebuttal resolved your weaknesses, and if so, to reflect this in your scores. Otherwise, we are happy to continue the discussion!
>
> Our rebuttal explains how modal aphasia explicitly concerns multimodal models with a *shared representation space*. This should resolve the soundness concerns:
>
> 1. Our findings are valid without knowing the explicit internals of (Chat)GPT-5.
> 2. Our controlled experiments are independent of how the open-weight models are initially pretrained.
> 3. Regarding modal aphasia, image generation and image understanding are very similar tasks, and there is no obvious reason why doing both should significantly change our findings.
>
> We further describe how we designed our safety case-study to be robust against prompt rephrasings, and that it serves as an illustrative example (not a general claim).

---

### Official Review · Reviewer_uwWN · 2025-11-05

**Soundness:** 3
**Presentation:** 3
**Contribution:** 3
**Rating:** 6
**Confidence:** 2

**Summary:**

The paper first points out that current unified multimodal models exhibit a systematic dissociation, termed modal aphasia, where they can generate highly accurate visual content but fail to access the same knowledge through text queries. To address this, the paper proposes a series of controlled experiments using open-weight unified models to analyze how this dissociation emerges across architectures and training setups. The study aims to reveal the inherent limitations of current multimodal knowledge transfer mechanisms and highlight the potential safety risks of modality-specific alignment, providing insights for developing models with genuinely unified cross-modal understanding.

**Strengths:**

* The paper introduces the new concept of modal aphasia, identifying a systematic dissociation between visual and textual understanding in unified multimodal models. This is an original and theoretically significant contribution that reframes existing assumptions about cross-modal knowledge transfer.

* The authors demonstrate modal aphasia not only in frontier models such as ChatGPT-5, but also in controlled experiments with open-weight models (Janus-Pro, Harmon). This dual-level validation strengthens the robustness and generality of the findings.

* The authors release code, data, and detailed experimental procedures, enhancing transparency and reproducibility. Their unified rubric-based evaluation also provides a standardized way to measure multimodal consistency.

**Weaknesses:**

* The paper successfully identifies modal aphasia as a systematic failure of multimodal models, but it does not offer a clear theoretical explanation or model-level mechanism to account for this behavior. The contribution remains largely descriptive rather than explanatory.

* Most experiments are conducted on controlled synthetic datasets such as fictional faces and geometric patterns. While these setups enable variable control, they do not demonstrate whether modal aphasia persists in realistic multimodal tasks such as captioning or retrieval. This limits the external validity of the findings.

* Although the authors link modal aphasia to potential safety risks, the provided evidence is based on a single case study involving a narrow concept category. The safety implications would be more convincing if supported by broader empirical evaluation across multiple harmful concept types or adversarial prompt settings.

* The conclusion suggests that allowing models to visualize concepts during reasoning may mitigate modal aphasia, but the paper does not propose any concrete implementation or experimental verification of this idea. This limits the practical contribution of the work.

I would like to discuss these points with the authors during the rebuttal stage and will adjust my score based on their responses and the feedback from other reviewers.

**Questions:**

See weaknesses.

---

> ### Author Response · Authors · 2025-11-20
>
> We thank the reviewer for recognizing the novelty and strength of our findings, and we appreciate their constructive suggestions.
>
> > [W1] The paper successfully identifies modal aphasia as a systematic failure of multimodal models, but it does not offer a clear theoretical explanation or model-level mechanism to account for this behavior.
>
> We agree that a theoretical explanation of modal aphasia would be useful. But this requires analyzing the learning behavior of billion-parameter models with multiple input and output modalities, for which we do not have the theoretical tools yet.
> We thus decided to focus on a rigorous empirical study.
>
> > [W2] Most experiments are conducted on controlled synthetic datasets [...]. While these setups enable variable control, they do not demonstrate whether modal aphasia persists in realistic multimodal tasks such as captioning or retrieval.
>
> Modal aphasia concerns how different modalities can access memorized knowledge. This certainly influences real-world tasks.
>
> However, most such tasks are also affected by other phenomena. For example, VLMs might ignore information in *input* images during VQA/captioning tasks [1]. This is independent of whether the model can *access memorized knowledge*.
>
> We hence deliberately study modal aphasia in controlled settings where we can rule out other factors, thereby strengthening our findings.
>
> > [W3] Although the authors link modal aphasia to potential safety risks, the provided evidence is based on a single case study involving a narrow concept category. The safety implications would be more convincing if supported by broader empirical evaluation across multiple harmful concept types or adversarial prompt settings.
>
> Our safety case-study is an illustrative example of how modal aphasia might affect a system's safeguards. Because real-world safeguards are complex (and often unknown), we deliberately keep the case-study simple for the sake of illustration.
>
> In particular, we do not make a general claim that the case-study directly reflects practice. However, the example concept is not cherry-picked or special, and we expect similar outcomes with different harmful concept types.
>
> > [W4] The conclusion suggests that allowing models to visualize concepts during reasoning may mitigate modal aphasia, but the paper does not propose any concrete implementation or experimental verification of this idea.
>
> The goal of our work is establishing modal aphasia as a systematic phenomenon, not mitigating it. We thus only conjecture image generation in CoT/thinking as a potential solution.
>
> In particular, we believe simply developing the proposed experiment consitutes its own future work:
> Almost all open-weight unified models are *not* trained for CoT reasoning with images; only a single architecture explicitly uses image generation in reasoning traces [2], but only for text-to-image tasks.
>
> Therefore, a concrete implementation or experimental verification requires a novel post-training method, which is highly non-trivial.
>
> We would like to ask the reviewer if they pictured a specific experiment that would convince them while being feasible in the scope of our work.
>
>
> [1] Liu et al., Paying More Attention to Image: A Training-Free Method for Alleviating Hallucination in LVLMs. In ECCV, 2024.
>
> [2] Chern et al., Thinking with Generated Images. arXiv preprint arXiv:2505.22525, 2025.

---

> ### Author Response · Authors · 2025-11-28
>
> Dear Reviewer uwWN,
>
> We would like to kindly ask you to let us know whether our rebuttal resolved your weaknesses, and if so, to reflect this in your scores. Otherwise, we are happy to continue the discussion!
>
> In summary: We discuss how the scope of our controlled experiments and safety case-study are not significantly limiting the contributions of our work. Moreover, we highlight why we think theoretical/model-level explanations or a concrete implementation of a mitigation are not currently feasible.

---

### Official Review · Reviewer_y19N · 2025-11-07

**Soundness:** 3
**Presentation:** 4
**Contribution:** 4
**Rating:** 8
**Confidence:** 4

**Summary:**

This paper investigates a problem in MLLMs called Modal Aphasia, a phenomenon that MLLMs can accurately memorize concepts visually but fail to articulate them in writing. This work starts with a case study of poster generation with GPT5 to reveal this problem. And then created two synthetic datasets to further validate the existence of this issue. Related AI safety concerns are raised with a case study.

**Strengths:**

S1: This paper is clearly written and easy to follow

S2: The found problem of modality imbalance in image/text generation fidelity is of great importance, and the naming is fun and accurate.

S3: The experiments to quantify and validate modal aphasia are well designed

**Weaknesses:**

W1: This work focuses on the modality imbalance problem of MLLMs in image/text generation. There are related studies/benchmarks in image/text understanding about modality imbalance in VLMs, which are worth discussing to better position this work.

W2: Since GPT5 is a proprietary model, there are rumors that its image generation is routed through another “sub-model” of GPT5. If so, the modality imbalance problem in image/text generation is kind of expected because of such a mismatch. I would like to know the authors’ thoughts on this.

**Questions:**

Please refer to W2.

---

> ### Author Response · Authors · 2025-11-20
>
> We thank the reviewer for raising important points and are happy to hear that they found our work important and rigorous.
>
> > [W1] This work focuses on the modality imbalance problem of MLLMs in image/text generation. There are related studies/benchmarks in image/text understanding about modality imbalance in VLMs, which are worth discussing to better position this work.
>
> We thank the reviewers for noting the apparent similarity between modality imbalance and modal aphasia; we add a discussion in the updated version of the paper.
>
> Although similar at a glance, modal aphasia is a novel phenomenon that is mostly orthogonal to existing work on modality imbalance.
>
> First, for multimodal classifiers, modality imbalance is typically attributed to the generalization speed of different modalities [1], one modality dominating the loss [2], or the distribution of modality-specific representation spaces [3]. Our controlled study eliminates all those effects by training only the LLM backbone (which has a single unified representation space).
>
> Second, modality imbalance for frontier models (e.g., [4]) means that *information in inputs* to one modality (typically text) gets preferred over inputs in other modalities. Modal aphasia concerns *memorization of knowledge* during training, and finds that certain modalities (here: text) can fail to access/express this *learned information*.
>
> > [W2] Since GPT5 is a proprietary model, there are rumors that its image generation is routed through another “sub-model” of GPT5. If so, the modality imbalance problem in image/text generation is kind of expected because of such a mismatch.
>
> We kindly refer the reviewer to our general response for a detailed answer. In short, even if the final image generation is indeed performed by some image decoder sub-model, this is likely done based on representations that can be decoded both into images and text. Yet, unexpectedly, the text modality fails to access the knowledge in those representations.
>
> [1] Wang et al., What Makes Training Multi-Modal Classification Networks Hard?. In CVPR, 2020.
>
> [2] Peng et al., Balanced Multimodal Learning via On-the-fly Gradient Modulation. In CVPR, 2022.
>
> [3] Ma et al., Revisit Modality Imbalance at the Decision Layer. arXiv preprint arXiv:2510.14411, 2025.
>
> [4] Liu et al., Paying More Attention to Image: A Training-Free Method for Alleviating Hallucination in LVLMs. In ECCV, 2024.

---

> ### Author Response · Authors · 2025-11-28
>
> Dear Reviewer y19N,
>
> We would like to kindly ask you to let us know whether our rebuttal resolved your questions. Otherwise, we are happy to continue the discussion!
>
> In short, we added a discussion on the differences between modality imbalance and modal aphasia, and we discuss how our findings are sound despite not knowing the specific internals of (Chat)GPT-5.

---

### Official Review · Reviewer_uNr3 · 2025-11-07

**Soundness:** 3
**Presentation:** 4
**Contribution:** 3
**Rating:** 6
**Confidence:** 3

**Summary:**

This paper presents an interesting phenomenon called modal aphasia, where leading unified multimodal models can faithfully generate almost perfect visual outputs but fail to describe details and concepts verbally. The authors demonstrate modal aphasia through observational experiments on GPT-5 and controlled fine-tuning on open-weight models. They also conduct a targeted case study to demonstrate bypassing safeguards by exploiting the modal aphasia.

**Strengths:**

This paper is well written. It starts with a clear motivation that derives from our daily interactions with commercial multimodal models. Then, the authors employ an interesting study of generating visually vs. describing verbally upon movie posters in frontier models, showing the prevalence of the modal aphasia. Beyond observational experiments, the authors design crisp synthetic data and fine-tuning experiments to show that modal aphasia can stem from more than just naive image memorization, but a systematic discrepancy of knowledge and concept understanding across modalities. Finally, they exploit a harmful use case if modal aphasia is not properly addressed. Overall, this paper is coherent and a joy to read.

**Weaknesses:**

1. The controlled experiments on open-source models only examine two open-source image generators. Also, the scale of test data is relatively small (below 200).

2. In the controlled fine-tuning for tracing the origin of modal aphasia, activating only the LLM backbone while freezing other components may not reflect real-world training.

**Questions:**

See Weaknesses. Also,
1. I am curious to see what happens with chain-of-thought prompting in these unified multimodal models? For example, instead of describing features in text independently, what will happen if the model is explicitly asked to "visualize then describe"?

2. The definition of modal aphasia sounds relevant to modality imbalance, which has already been well-identified and mechanistically understood in quite a few previous works. How do the authors view the similarity and difference compared with modality imbalance in general VLMs or multimodal models?

In general, the paper’s claims and findings are self-contained, and weaknesses are relatively minor. I’d be happy to raise the score if the questions are adequately addressed.

---

> ### Author Response · Authors · 2025-11-20
>
> We thank the reviewer for their insightful comments and are happy that they enjoyed our work.
>
> > [W1] The controlled experiments on open-source models only examine two open-source image generators.
>
> There exist only few open-weight unified models that i) train/align modalities jointly, ii) are capable enough for our experiments (e.g., have strong LLM backbones), and iii) are general-purpose (e.g., do not focus on only specific tasks such as image editing).
>
> Additionally, supporting a model requires non-trivial engineering efforts, because there are no general APIs for training unified architectures.
>
> We hence chose only two models but made sure they satisfy all aforementioned criteria and are diverse in terms of training and architecture paradigms.
>
>
> > [W1] Also, the scale of test data is relatively small (below 200).
>
> Since we investigate memorization of knowledge, the test set plays a minor role; it is used only to claim that models trained on synthetic abstract concepts generalize to unseen combinations of such concepts.
>
> Our fine-tuned models achieve between 74% and 95% test accuracy when generating unseen combinations. Since the random-guessing baseline is around 19%, we argue that our claim is statistically sound even on the small test set.
>
> Lastly, even if that weren't the case, the findings of modal aphasia for abstract synthetic concepts are independent of how well a model performs on the test set.
>
> > [W2] In the controlled fine-tuning for tracing the origin of modal aphasia, activating only the LLM backbone while freezing other components may not reflect real-world training.
>
> We kindly refer the reviewer to our general response. In short, we agree that the setup does not reflect the real world. But our setup is designed specifically to ensure that memorization of knowledge can happen only in the unified representation space of the backbone. Thereby, we strengthen our findings, as we rule out trivial cases of modal aphasia.
>
> > [Q1] I am curious to see what happens with chain-of-thought prompting in these unified multimodal models? For example, instead of describing features in text independently, what will happen if the model is explicitly asked to "visualize then describe"?
>
> This is an excellent question! We are currently running the suggested experiment and will include the outcome in the updated version of our paper. For now, we expect no significant improvements; we conjecture that training-time interventions are necessary to mitigate modal aphasia.
>
> > [Q2] How do the authors view the similarity and difference compared with modality imbalance in general VLMs or multimodal models?
>
> We thank the reviewers for noting the apparent similarity between modality imbalance and modal aphasia; we add a discussion in the updated version of the paper.
>
> Although similar at a glance, modality imbalance and modal aphasia are different phenomena.
>
> First, for multimodal classifiers, modality imbalance is typically attributed to the generalization speed of different modalities [1], one modality dominating the loss [2], or the distribution of modality-specific representation spaces [3]. Our controlled study eliminates all those effects by training only the LLM backbone (which has a single unified representation space).
>
> Second, modality imbalance for frontier models (e.g., [4]) means that *information in inputs* to one modality (typically text) gets preferred over inputs in other modalities. Modal aphasia concerns *memorization of knowledge* during training, and finds that certain modalities (here: text) can fail to access/express this *learned information*.
>
> Our work hence introduces a novel phenomenon that is mostly orthogonal to existing work on modalitiy imbalance.
>
> [1] Wang et al., What Makes Training Multi-Modal Classification Networks Hard?. In CVPR, 2020.
>
> [2] Peng et al., Balanced Multimodal Learning via On-the-fly Gradient Modulation. In CVPR, 2022.
>
> [3] Ma et al., Revisit Modality Imbalance at the Decision Layer. arXiv preprint arXiv:2510.14411, 2025.
>
> [4] Liu et al., Paying More Attention to Image: A Training-Free Method for Alleviating Hallucination in LVLMs. In ECCV, 2024.

---

> ### Author Response · Authors · 2025-11-28
>
> Dear Reviewer uNr3,
>
> We would like to kindly ask you to let us know whether our rebuttal resolved your weaknesses and questions, and if so, to reflect this in your scores. Otherwise, we are happy to continue the discussion!
>
> In short: Our rebuttal explains how the points in the weaknesses do not influence the significance of our findings. We further added the experiment from Q1 (prompting a model to "visualize then describe" does not suffice to resolve modal aphasia) and a discussion on the differences between modality imbalance and modal aphasia.

---

### Author Response · Authors · 2025-11-20
**General response to all reviewers**

We thank all reviewers for their constructive feedback! We are happy to hear that reviewers found our findings of modal aphasia "original and theoretically significant" (uwWN), "of great importance" (y19N), "interesting and potentially surprising" (tN39), and supported by "robust" and "well designed" experiments (uwWN, y19N).

We are currently incorporating the feedback into our paper and will upload a revised version by November 24th.

We address common inputs raised by multiple reviewers below:

> 1. For experiments on closed-source models such as GPT-5, we cannot know if the model is truly unified, or if it uses a sub-model to generate images (y19N, Zj2z, tN39)

We agree that we cannot know this for certain. However, there is significant evidence that OpenAI's recent models use a *joint representation space* for images and text (in contrast to models before the current GPT-4o that indeed called an external image generator with text prompts).

For instance, the [official release for GPT-4o image generation](https://openai.com/index/introducing-4o-image-generation/) mentions "We trained our models on the joint distribution of online images and text", "Because image generation is now native to GPT‑4o, you can refine images through natural conversation. GPT‑4o can build upon images and text in chat context, ensuring consistency throughout", or, "Native image generation enables 4o to link its knowledge between text and images".

GPT-5 exhibits similar capabilities (e.g., accurate image editing across messages) that are only feasible if images and text are represented in a unified space.

Thus, even if the final image generation is indeed performed by some separate image decoder model, this is likely done based on representations that can be decoded into both images and text.

Nevertheless, we do not have proof that this is how GPT-5 works
(and it is possible that the image decoder itself has memorized certain concepts). This is precisely why we also include a controlled study on open models where we can control exactly what happens and guarantee that memorization happens in the shared representation space.

> 2. For open models, why do you train only the LLM backbone? (uNr3, tN39)

As mentioned above, the point of our experiments with open models is to precisely control where the memorization occurs in the model, and guarantee that this is happening in weights that are shared across modalities, rather than in the parameters of a modality-specific encoder or decoder.

If we were to finetune all parameters, it is possible that memorization would occur in the image decoder, in which case modal aphasia would be expected and unavoidable.

By freezing all modality-specific components and fine-tuning only the shared LLM backbone, we ensure that any visual knowledge that is memorized can (in principle) also be decoded through text. The fact that this doesn't happen shows that a non-trivial form of modal aphasia must be present.

> 3. The Safety case-study is not evidence of modal aphasia. The effects could be due to poor robustnes to prompt manipulations. (Zj2z, tN39)

While we agree that we cannot guarantee that the effects are due to modal aphasia, we take multiple steps to isolate spurious effects due to changes in prompts.

Our safety fine-tuning is simplistic due to practical constraints, but we did aim to make it robust to different phrasings. The training and validation prompts include a diverse set of phrasings, including adversarial ones (e.g., "bar3 f00t illustration."), and the resulting models are reasonably robust against such rephrasings.
Moreover, we ensured that the phrase "secondary balance units" is not used online (we found exactly 2 Google search results at the time of writing, none related to feet).

We thus believe it is unlikely that our results are just due to prompt engineering. Instead, the outcome of the study aligns with modal aphasia: the fine-tuned Janus-Pro models learn the visual concept of feet. But since the text modality does not "understand" this representation, it learns to simply refuse for *prompts mentioning the word "feet"* instead of refusing to generate *images containing the concept of feet*.

If the reviewers have suggestions for alternative experiments that could be more convincing, we would be happy to try them out!

---

### Author Response · Authors · 2025-11-24
**Updated paper is posted**

As promised, we uploaded a revised version of the paper that incorporates the reviewer's feedback. The following lists all the changes and additions, including where to find them.

1. An ablation of the real-world experiments from Sec. 3, where we explicitly ask ChatGPT-5 to "visualize" movie posters before describing them. (Appendix A.3)
2. Additional related work comparing modal aphasia to modality imbalance and evaluations of visual/spatial commonsense reasoning. (Appendix D)
3. All bar plots of our controlled experiments now contain random guessing baselines. (Figures 4, 5, 9, 10)
4. Benchmark results now also include the base (pretrained) Janus-Pro and Harmon models. (Table 5)
5. General improvements of the writing and presentation, in particular
    1. A better explanation of why our real-world findings are reasonable despite not knowing the internals of ChatGPT-5. (Appendix A.1)
    2. An explanation of why we do not repeat the real-world experiments with the base (zero-shot) Janus-Pro and Harmon models. (Appendix A.1)
    3. Clarification of ChatGPT-5 vs. GPT-5. (Appendix A.1)
    4. Removal of all accidental claims that models train all modalities jointly "from scratch".

---

### Meta-Review · Area_Chair_87XQ · 2026-01-11

**Summary:**

The main concerns from the reviewers are following:

- Reviewer **uNr3**:
  - **W1**: Discussing the relevance to researches on modality imbalance.

- Reviewer **y19N**:
  - **W1**: Discussing the relevance to researches on modality imbalance.
  - **W2**: The soundness of the results under GPT-5.

- Reviewer **uwWN**:
  - **W1**: The contribution remains largely descriptive rather than explanatory.
  - **W2**: Most experiments are conducted on controlled synthetic datasets such as fictional faces and geometric patterns.
  - **W3**: Although the authors link modal aphasia to potential safety risks, the provided evidence is based on a single case study involving a narrow concept category.
  - **W4**: The conclusion suggests that allowing models to visualize concepts during reasoning may mitigate modal aphasia, but the paper does not propose any concrete implementation or experimental verification of this idea.

 - Reviewer **Zj2z**:
    - **W1**: To obtain the multimodal models in the experiments, jointly training the model on both image generation and image captioning is more proper than training on image generation only.
    - **W2**: Issue of conflating modal aphasia with a prompt engineering loophole.
    - **W3**: The assumption that modern unified architectures train image and language jointly "from scratch" is invalid.

 - Reviewer **tN39**:
    - **W1**: The soundness of the results under GPT-5.
    - **W2**: It is in doubt that performance when fine-tuning on a new visual concept would be comparable to zero-shot evaluation of the pretrained model’s knowledge of visual concepts.
    - **W3**: It is in doubt that if the effect demonstrated in Sec 5 is due to modal aphasia.

**Reviewer Concerns:**

- **Concerns addressed in the rebuttal:**
  - The author responses provide detailed clarifications on reviewers' concerns, in special the ones from the negative reviewers. The authors also admit that some designs are not absolutely rigorous (**W3** of Reviewer **uwWN**, **W2, W3** of Reviewer **tN39**, **W2** of Reviewer **Zj2z**), while I think these shortages are acceptable compared with the merits.

- **Concerns remained outstanding:**
  - No outstanding concerns remaining after author responses.

**Reviewer Scores:**

As discussed in the Reviewer Concerns, most points are indeed addressed by the author responses. From the viewpoint of the negative reviewers, since the paper proposes an interesting problem and provides thought-provoking results, I would treat the remaining drawbacks as minor ones and would not insist rejection.

---

### Decision · Program_Chairs · 2026-01-26

Accept (Poster)